# Research on the Elastic Loss Characteristics of Acoustic Echoes from Underwater Corner Reflector

**DOI:** 10.3390/s25123776

**Published:** 2025-06-17

**Authors:** Yi Luo, Dawei Xiao, Jingzhuo Zhang, Zuqiu Li

**Affiliations:** Naval University of Engineering, Wuhan 430033, China; luomx@163.com (Y.L.); 13982402893@163.com (Z.L.)

**Keywords:** subsurface corner reflector, concave structure, rigidity, elasticity loss, echo characteristics, pool experiments

## Abstract

The underwater corner reflector is a “concave” elastic structure, and its acoustic echo exhibits large elastic loss, which affects its practical use. To study the acoustic echo elastic loss characteristics of underwater corner reflectors, based on the characteristics of small concave elastic structures of underwater corner reflectors, theoretical calculations were performed using the method of a combination of finite element and boundary element. Taking the underwater rigid corner reflector as the benchmark, the acoustic echo differences between similar types of underwater elastic corner reflectors were compared. The regular acoustic echo elastic loss of underwater corner reflectors was analyzed, and verified through pool experiments. The results show that, whether single-grid or multi-grid corner reflector, the actual acoustic echoes of underwater corner reflectors conform to the characteristics of elastic bodies, which differ significantly from rigid bodies and exhibit obvious elastic loss. The elastic loss mainly manifests as reduced target strength (TS), narrower directional pattern width, and poorer frequency stability of target strength, which is detrimental to practical use. This study provides assistance in proposing targeted methods to suppress elastic loss.

## 1. Introduction

A corner reflector is a concave structure with vertical surfaces (as shown in Figure 1 and Figure 2), which is widely used for electromagnetic reflection on the ground [1,2,3,4] and electromagnetic marking [5,6,7,8]. It has advantages such as strong echo capability, simple structure, ease of use, and strong environmental adaptability [9,10]. The literature [11,12,13,14] proposes the idea of utilizing the advantages of a corner reflector, as well as for underwater acoustic marking and other applications.

To realize this application idea, the underwater corner reflector must have excellent acoustic reflection capabilities.

However, the electromagnetic wave reflection of corner reflectors is not entirely the same as underwater acoustic reflection, and actual underwater corner reflectors are not ideal rigid bodies. For example, although common metallic corner reflectors have important advantages, such as good structural stability, they are essentially concave elastic structures composed of thin plates. From the perspective of the acoustic echo process, in most incident directions, the incident wave not only produces geometric and elastic scattering waves on the direct incident surface, but also reaches other reflecting surfaces, causing various scattering waves, multiple geometric and elastic scattering, and interactions, which reduce the strength of the reverse echo and significantly increase elastic loss. Only by studying their elastic loss characteristics can we propose targeted suppression methods to make them truly strong underwater echo structures with practical value. However, there is currently little research on the acoustic echo elastic loss patterns of underwater corner reflectors.

The theoretical research on acoustic scattering from complex underwater structures primarily employs numerical calculation methods [15,16,17]. Scholars have proposed various methods according to different underwater structures and research focuses.

The physical acoustics method can solve the scattered sound field of convex targets with arbitrary shapes, and the calculation process is simple, but it is only suitable for high-frequency calculations. The panel element method, based on the physical acoustics method, improves the calculation efficiency and is suitable for high-frequency calculations of rigid structures, but it is difficult to calculate elastic structures. The T-matrix method is suitable for convex smooth surface targets. The advantage of the finite element method [18] is its high calculation accuracy, which makes it suitable for any complex-shaped target. However, its large calculation amount makes it impossible to accurately solve the sound scattering problem in infinite domains. The boundary element method [19] is suitable for any complex-shaped target and can solve the sound scattering problem in infinite domains, but it is difficult to calculate the high-frequency scattering of large-scale targets. Using the combination of finite element and boundary element methods [19,20] to solve the sound scattering of underwater elastic structures can leverage the advantages of both methods, making it suitable for complex-shaped targets with high accuracy.

Considering that underwater corner reflectors are generally made with thin plates, and are small in size and concave, the combination of finite element and boundary element methods is an effective theoretical approach for the study of the acoustic scattering problem of such small, concave, multi-faceted thin plate structures.

Therefore, this paper focuses on the structural characteristics of underwater corner reflectors, using the method of finite element combined with boundary element for theoretical calculations, and the acoustic echo differences between underwater rigid and elastic corner reflectors are compared; the influencing factors and magnitude of the elastic loss of acoustic echoes from underwater corner reflectors are studied with pool experiments for verification.

## 2. Calculation Method

To solve the acoustic scattering problem of underwater targets using the finite element method (FEM) combined with the boundary element method (BEM), two methods can be adopted: FEM combined with the direct boundary element method (DBEM) and FEM combined with the indirect boundary element method (IBEM). DBEM requires the acoustic field domain to be non-open, meaning the surface acoustic grid is closed. Its applicable scenarios include purely internal closed acoustic problems or external acoustic problems. The surface acoustic grid of IBEM can be either closed or non-closed. Essentially, there is no fundamental difference between these two methods; the difference lies in the type of acoustic grid. When using DBEM, the elastic structure is only in contact with the fluid on one side; when using IBEM, the elastic structure can be in contact with the fluid on one side, or on both sides simultaneously with the same fluid.

In the acoustic–structure interaction system, the acoustic boundary element equation (acoustic BE Model) is used to calculate the acoustic response of the boundary surface, while the structural finite element equation (structural FEM Model) is used to calculate the surface displacement of the elastic structure. The combination of finite element and direct boundary element is a classical theory, which is no longer described. The finite element combined with indirect boundary element model is given below.

Different from direct boundary elements, the mesh of indirect boundary elements is open, so the calculated sound field is located on both sides of the mesh.

### 2.1. Acoustic Boundary Element Equation

The single-layer potential (sound pressure gradient difference) function $\overset{\wedge }{\sigma }$ and double-layer potential (sound pressure difference) function $\overset{\wedge }{\mu }$ on the boundary surface are represented as(1)$$\overset{\wedge }{\sigma }(\bar{r_{a}})=\left[N_{\sigma }\right]\cdot \left\{{\overset{\wedge }{\sigma }}_{i}\right\},\bar{r_{a}}\in \Omega_{\sigma }$$(2)$$\overset{\wedge }{\mu }(\bar{r_{a}})=\left[N_{\mu }\right]\cdot \left\{{\overset{\wedge }{\mu }}_{i}\right\},\bar{r_{a}}\in \Omega_{\mu }$$

In the Formula (1), $\left[N_{\sigma }\right]$ is the global shape function associated with the nodes on the boundary surface $\Omega_{\sigma }$ and $n_{\sigma }$; $\left[N_{\mu }\right]$ is the global shape function associated with the nodes on the boundary surface $\Omega_{\mu }$ and $n_{\mu }$; $\left\{{\overset{\wedge }{\sigma }}_{i}\right\}\text{,}\left\{{\overset{\wedge }{\mu }}_{i}\right\}$ represent the single-layer potential and double-layer potential vectors of the nodes, respectively.(3)$${\overset{\wedge }{\sigma }}_{i}=-j\rho_{0}\omega (v_{1i}-v_{2i})$$(4)$${\overset{\wedge }{\mu }}_{i}=p_{1i}-p_{2i}$$

In the formula, $\rho_{0}$ is the fluid density; ω is the circular frequency; $v_{1i}$ and $v_{2i}$ are the normal vibration velocities on both sides of the surface, respectively; and $p_{1i}$ and $p_{2i}$ are the sound pressure on both sides of the surface.

Without external sound source excitation, the indirect boundary element equation is expressed as(5)$$\left[\begin{array}{cc} B & C\\ C^{T} & D \end{array}\right]\cdot \left\{\begin{array}{c} {\overset{\wedge }{\sigma }}_{i}\\ {\overset{\wedge }{\mu }}_{i} \end{array}\right\}=\left\{\begin{array}{c} \overset{\approx }{f_{\sigma }}\\ \overset{\approx }{f_{\mu }} \end{array}\right\}$$

In the formula, $\left[B\right]\text{,}\;\left[C\right]\text{,}\;\left[D\right]$ represents the influence matrix; $\left\{\overset{\approx }{f_{\sigma }}\right\}\text{,}\;\left\{\overset{\approx }{f_{\mu }}\right\}$ represent the excitation vector.

### 2.2. Structural Finite Element Equation


(6)
$$\left(\left[K_{s}\right]+j\omega \left[C_{s}\right]-\omega^{2}\left[M_{s}\right]\right)\cdot \left\{w_{i}\right\}=\left\{F_{s}\right\}$$


In the formula, $\left[K_{s}\right]\text{,}\;\left[C_{s}\right]\text{,}\;\left[M_{s}\right]$ express the stiffness, damping and mass matrices of the structure, respectively; $\left\{w_{i}\right\}$ is the node displacement vector; the vector $\left\{F_{s}\right\}$ contains constraint degrees of freedom, forces applied to the boundary surface of the structure, and external loads applied perpendicular to the surface.

### 2.3. Coupled Equations

The boundary element nodes consist of nodes located on the coupled surface $\Omega_{s}$ and nodes on the remaining boundary surfaces $\Omega_{a}\backslash \Omega_{s}$. At this point, the single-layer potential on the boundary surface $\overset{\wedge }{\sigma }$ is 0, and the double-layer potential is(7)$$\overset{\wedge }{\mu }(\bar{r_{a}})=\left[N_{\mu 1}\right]\left\{{\overset{\wedge }{\mu }}_{i1}\right\}+\left[N_{\mu 2}\right]\left\{{\overset{\wedge }{\mu }}_{i2}\right\}\;,\;\bar{r_{a}}\in \Omega_{a}$$

In the formula, $\left[N_{\mu 1}\right]$ represents the global shape function related to the coupled surface degrees of freedom in the vector $\left\{{\overset{\wedge }{\mu }}_{i1}\right\}$; $\left[N_{\mu 2}\right]$ represents the global shape function related to the remaining boundary surface pressure degrees of freedom in the vector $\left\{{\overset{\wedge }{\mu }}_{i2}\right\}$; $\left\{{\overset{\wedge }{\mu }}_{i1}\right\}$ and $\left\{{\overset{\wedge }{\mu }}_{i2}\right\}$ are double-layer potential vectors corresponding to surface nodes, respectively.

The sound pressure difference at the coupled interface is regarded as the applied normal load. At this point, the structural finite element equation can be expressed as(8)$$\left(\left[K_{s}\right]+j\omega \left[C_{s}\right]-\omega^{2}\left[M_{s}\right]\right)\cdot \left\{w_{i}\right\}+\left[L_{c}\right]\cdot \left\{{\overset{\wedge }{\mu }}_{i1}\right\}=\left\{F_{s}\right\}$$

The coupling matrix $\left[L_{c}\right]$ is(9)$$\left[L_{c}\right]=\sum_{e=1}^{n_{se}}\left(\int_{\Omega_{se}}\left({\left[N_{s}\right]}^{T}\cdot \left\{n^{e}\right\}\cdot \left[N_{\mu 1}\right]\right)\cdot d\Omega \right)$$

In the formula, $n_{se}$ represents the number of elements on the coupled surface $\Omega_{se}$; $n^{e}$ is the normal direction of the cell; $\left[N_{s}\right]$ is the global shape function matrix related to $n_{s}$ unconstrained degrees of freedom. Since the single-layer potential $\overset{\wedge }{\sigma }$ is 0, the indirect boundary element Equation (5) can be expressed as(10)$$\left[\begin{array}{cc} D_{11} & D_{12}\\ D_{21} & D_{22} \end{array}\right]\cdot \left\{\begin{array}{c} {\overset{\wedge }{\mu }}_{i1}\\ {\overset{\wedge }{\mu }}_{i2} \end{array}\right\}=\left\{\begin{array}{c} \overset{\approx }{f_{\mu 1}}\\ \overset{\approx }{f_{\mu 2}} \end{array}\right\}+\left\{\begin{array}{c} \overset{\approx }{f_{\mu s}}\\ 0 \end{array}\right\}$$

The coefficient vectors $\left\{\overset{\approx }{f_{\mu 1}}\right\}\text{,}\;\left\{\overset{\approx }{f_{\mu 2}}\right\}$ are, respectively,(11)$$\left\{\overset{\approx }{f_{\mu 1}}\right\}=\int_{\Omega_{a}\backslash \Omega_{s}}j\rho_{0}\omega {\bar{v}}_{n2}{\left[N_{\mu 1}\right]}^{T}\cdot d\Omega$$(12)$$\left\{\overset{\approx }{f_{\mu 2}}\right\}=\int_{\Omega_{a}\backslash \Omega_{s}}j\rho_{0}\omega {\bar{v}}_{n2}{\left[N_{\mu 2}\right]}^{T}\cdot d\Omega$$

${\bar{v}}_{n2}$ is the normal velocity $\Omega_{a}\backslash \Omega_{s}$ on the boundary surface.

In the case of complete coupling between structure and fluid, the boundary element Formula (10) and the finite element Equation (8) are combined to obtain the coupled equation system:(13)$$\left[\begin{array}{cc} K_{s}+j\omega C_{s}-\omega^{2}M_{s} & L_{c}\\ L_{c}^{T} & \frac{D}{\rho_{0}\omega^{2}} \end{array}\right]\left\{\begin{array}{l} w_{i}\\ {\overset{\wedge }{\mu }}_{i} \end{array}\right\}=\left\{\begin{array}{c} F_{s}\\ F_{a} \end{array}\right\}$$

In the formula,(14)$$\left\{F_{a}\right\}=-\sum_{e=1}^{n_{se}}\left(\int_{\Omega_{se}}\left({\left[N_{\mu }\right]}^{T}\cdot {\left\{n^{e}\right\}}^{T}\cdot \left[N_{w}\right]\cdot \left\{{\bar{w}}_{i}\right\}\right)\cdot d\Omega \right)$$

In the Formula (14), $\left[N_{w}\right]$ is the global shape function matrix related to $n_{w}$ degrees of freedom.

The sound pressure and intensity at any point in space are solved by the finite element method combined with the boundary element method, and the corresponding TS is obtained by using the following formula:(15)$$TS=20\mathrm{lg}\frac{{\left.p_{r}\right|}_{r=1}}{p_{i}}=10\mathrm{lg}\frac{{\left.I_{r}\right|}_{r=1}}{I_{i}}$$

In the Formula (15), ${\left.p_{r}\right|}_{r=1}$ and ${\left.I_{r}\right|}_{r=1}$ are the reflected wave sound pressure and sound intensity 1 m away from the target acoustic center, respectively; $p_{i}$ and $I_{i}$ are the incident wave sound pressure and sound intensity, respectively.

## 3. Numerical Simulation and Analysis

### 3.1. Simulation Software and Calculation Process

Theoretical calculations were performed using acoustic simulation software. Virtual Lab is an advanced acoustic simulation software [21] that can calculate the acoustic response of an object, such as sound pressure, sound intensity, and sound power. Due to the relatively weak modeling capabilities of Virtual Lab software (LMS Virtual.Lab 13.0), there are difficulties in establishing complex target models. Therefore, this paper first uses ANSYS 2021R1software to establish the mesh model, which is then imported into Virtual Lab for simulation calculations. The main process of simulation using the aforementioned software in this paper is shown in Figure 3.

In order to ensure the calculation accuracy, the shell-181 type element is used to mesh the model in ANSYS software, and the mesh density is controlled accordingly. The denser the mesh, the higher the calculation accuracy, but the larger the amount of calculation. Generally, the accuracy requirements can be met by ensuring that there are at least six element grids within the unit wavelength.

In the LMS Virtual lab software, the incident sound wave is defined as a plane wave, the amplitude is 1 Pa, the sound source is 100 m away from the center of the target, and the field point is located at the sound source (transceiver combined), so that the calculation results meet the far-field conditions. The calculation of the sound field reflected by the corner reflector is carried out in two steps. The first step is to calculate the whole sound field, including the incident sound field and the scattered sound field; the second step is to calculate the scattered sound field. Finally, by calculating the scattered sound field, the sound pressure at the field point is extracted, and then the calculation is completed by using the Formula (15) of the TS.

For the rigid corner reflector, it is assumed to be composed of a rigid and smooth flat plate, with no energy loss during the reflection process. When the corner reflector is fully immersed in water, the surface of the reflector couples with the sound field in an infinite ideal fluid medium that is stationary. Its acoustic boundary conditions satisfy the equality of normal vibration velocities on the structure’s surface, which are all zero, and the continuity of surface sound pressure.

### 3.2. Acoustic Echo Elasticity Loss of Single-Grid Corner Reflector

#### 3.2.1. Echo Characteristics of Rigid Corner Reflectors

Firstly, the underwater corner reflector is assumed to be an ideal rigid body, and its echo sound field is simulated and calculated. Figure 4 is a schematic diagram of plane wave incidence on a single-grid corner reflector.

(1) Incident angle effect on rigid corner reflectors

The side length of the corner reflector is set to 1 m, with the thickness of 20 mm. The density of water is 1000 kg/m^3^, and the speed of sound in water is 1480 m/s. The incident sound wave is defined as plane wave with frequency of 15 kHz. The echoes (i.e., TS) of single-grid triangular, square, and circular corner reflectors are analyzed with the different incident angle, and the calculation results are shown in Figure 5 and Figure 6.

The following can be seen from Figure 5 and Figure 6:(a)When same sizes, the TS value of the square corner reflector > the circular > the triangular(b)The trend of TS variation with incident angle for the three types is basically consistent. When *φ* = 45° and *θ* ≈ 55°, the TS reaches its maximum, which are 22.7 dB, 20.2 dB, and 14.6 dB, respectively.

The spatial distribution of echo intensity varies among different corner reflectors. Using the calculation results from Figure 6, we obtain the scattering pattern of a single-grid corner reflector at an incident angle *θ* = 55° and *φ* = 0~90°, as shown in Figure 7.

As can be seen from Figure 7, the scattering pattern width of the triangular corner reflector is the largest, approximately 46°, while the scattering pattern widths of the square and circular corner reflectors are 39° and 30°, respectively.

(2) Incident frequency effect on rigid corner reflectors

Next, we analyze the relationship between the TS and incident frequency of the underwater corner reflector. With the incident wave angle set to θ=55°, and the frequency ranging from 10 kHz to 20 kHz, we calculate the variation in the maximum TS of three types of corner reflectors, as shown in Figure 8.

As can be seen from Figure 8; when the frequency of the incident wave increases; the TS of the rigid corner reflector first slowly increases and then tends to level off. This indicates that there is no significant frequency effect on the echo intensity of underwater rigid corner reflectors; especially in the high-frequency band

#### 3.2.2. Echo Characteristics of Elastic Corner Reflectors

Considering the corner reflector as a rigid body is merely an ideal scene, as actual corner reflectors are constructed from elastic materials, such as metal plates. Next, we analyze the echo characteristics of elastic corner reflectors. Taking the triangular metal corner reflector as an example for simulation analysis, we assume the material of the corner reflector to be the common steel plate (with a Young’s modulus of 2 × 10^11^ Pa, Poisson’s ratio of 0.3, and density of 7800 kg/m^3^); other parameters remain the same as before.

(1) Incident angle effect on elastic corner reflectors

Figure 9 illustrates the variation in TS of triangular corner reflector when the incident angle *φ* is 45° and *θ* ranges from 0 to 90°.

From Figure 9, we can see the following:(a)The echo characteristics of triangular corner reflectors fluctuate significantly with changes in incident angle, and the TS value increases with increasing frequency.(b)At 15 kHz, the TS reaches its minimum value at around 5°, and its maximum value at around 60°.

Figure 10 is the graph showing the variation of TS for the triangular corner reflector at incident angle *θ* = 55°, *φ* = 0~90°.

The following can be seen from the Figure 10:(a)The TS of the corner reflector is relatively large near *φ* = 0°, 90°, and 45°.(b)The curve representing the variation of TS of corner reflectors with incident angle exhibits significant fluctuations, and this variation becomes more remarkable as the frequency increases.

(2) Incident frequency effect on elastic corner reflectors

Figure 11 depicts the TS frequency response curve of the triangular corner reflector when the incident angle *θ* is 55° and *φ* is 45°.

The following can be seen from the Figure 11:(a)The TS of triangular corner reflectors exhibits significant variations in maximum and minimum values with frequency.(b)At low frequencies, the resonance peaks of the TS frequency response curve are denser. At high frequencies, the resonance peaks decrease, and the amplitude of peak-to-valley variations also decreases relatively.

The echo characteristics of single-grid square and circular corner reflectors are similar to those of triangular corner reflectors. Due to space limitations, we will not repeat them here.

#### 3.2.3. Elasticity Loss Analysis

Using underwater single-grid rigid corner reflectors as benchmark, we compared the echo characteristics of underwater single-grid elastic corner reflectors. Figure 12 and Figure 13 show the comparison results of underwater acoustic echoes between triangular steel plate corner reflector (made of elastic material) with side length of 1 m and plate thickness of 20 mm, and the rigid corner reflector of the same size.

It can be observed that compared to rigid corner reflectors of the same size, the underwater acoustic echo characteristics of elastic material corner reflectors exhibit significant differences, with evident elastic loss:(1)TS: In most incident directions, the TS of rigid corner reflectors is significantly greater than that of elastic corner reflectors. The average TS of rigid corner reflectors is 10.8 dB, while the average TS of elastic corner reflectors is only 2.5 dB.(2)Pattern width: The pattern width of the rigid corner reflector is significantly larger than that of an elastic corner reflector. The echo pattern width of the rigid corner reflector is approximately 46°, while that of the elastic corner reflector is approximately 15°.(3)Frequency stability of TS: When the frequency of the incident wave increases, the TS of the rigid corner reflector slowly increases and tends to be flat, indicating good frequency stability of the TS, whereas the TS of the elastic corner reflector fluctuates greatly with the frequency of the incident wave, indicating poor frequency stability of the TS.

The reason is the corner reflectors are essentially concave elastic structures composed of thin plates. When sound waves reach the underwater corner reflector, complex acoustic–vibration coupling occurs. From the perspective of the acoustic echo progress, in most incident directions, the incident wave not only generates geometric and elastic scattering waves on the direct incident surface, but also reaches other reflecting surfaces and induces various scattering waves, undergoing multiple geometric scattering, elastic scattering, and interactions, resulting in reduced intensity of the reverse echo and significant elastic loss.

Elastic loss is detrimental to the practical use of underwater corner reflectors. The low TS means that larger-sized corner reflectors must be used to simulate the acoustic reflection strength of underwater targets, and affect practicality. The narrow directional pattern width means that more corner reflectors must be used to simulate the directional characteristics of underwater target acoustic reflections, in order to avoid acoustic reflection blind spots. The poor frequency stability of TS is inconsistent with the generally stable TS of large-scale underwater targets.

### 3.3. Acoustic Echo Elasticity Loss of Multi-Grid Corner Reflector

#### 3.3.1. Echo Characteristics of Rigid Corner Reflectors

When the underwater corner reflector is used as an acoustic reflection device, it is often required to have a wide echo pattern.

Figure 14 is the schematic diagram of sound wave incidence on the eight-grid corner reflector, where φ is the angle between the projection of the incident sound wave on the *Oxy* plane and the *x*-axis, θ is the angle between the incident sound wave and the *z*-axis, and *h* is the thickness of the plate.

The following diagram Figure 15 is a simulation of the underwater echo sound field of an eight-grid triangular rigid corner reflector with side length of 1 m and thickness of 20 mm. The incident angle of the sound wave θ=55° is taken from 0° to 360°.

As can be seen from Figure 15, the eight-grid corner reflector can effectively increase the width of the echo pattern. Due to the symmetry of the corner reflector, the echo pattern has multiple axes of symmetry within the range of 0° to 360°.

The other echo properties of the eight-grid rigid corner reflector are the same as those of the single-grid corner reflector of the same type, and will not be repeated here.

#### 3.3.2. Echo Characteristics of Elastic Corner Reflectors

Taking the eight-grid triangular elastic corner reflector as an example, let us assume the material of the eight-grid corner reflector is common steel plate (with a Young’s modulus of 2 × 10^11^ Pa, a Poisson’s ratio of 0.3, and a density of 7800 kg/m^3^); other parameters remain the same as before.

The simulation results of the TS of corner reflectors under different incident frequencies and angles are shown in Figure 16.

From Figure 16, it can be see that

(1)When the incident angle of the sound wave is θ = 90°, φ = 0°~3°, φ = 15°~75°, and φ = 87°~90°, the TS increases with the increase of incident frequency; due to the symmetry of the reflector, the TS curve is axially symmetric along φ = 45°; TS reaches its maximum at φ = 0° and φ = 90°; around φ = 45°, the TS value remains basically stable within a range of about 45°, with the variation of no more than 10 dB.(2)Due to the complexity of the multi-grid corner reflector structure and the elasticity of the metal plate, more complex scattered sound waves will be generated. When the sound waves are incident at angles of φ = 3°~15° and φ = 75°~87°, the TS curve fluctuates greatly with weaker regularity, indicating a weaker sound reflection ability of the corner reflector.

#### 3.3.3. Elasticity Loss Analysis

Using the underwater eight-grid rigid corner reflector as the benchmark, we compared the echo characteristics of the underwater eight-grid elastic corner reflector. Figure 17 and Figure 18 show the comparison results of underwater acoustic echoes between the eight-grid triangular steel plate corner reflector (made of elastic material) with side length of 1 m and plate thickness of 20 mm, and the eight-grid rigid corner reflector of the same size.

It can be observed that, compared with rigid corner reflectors of the same size, multi-grid elastic corner reflectors exhibit significant differences in underwater acoustic echo characteristics, with evident elastic loss:(1)TS: In most incident directions, the TS of rigid corner reflectors is significantly greater than that of elastic corner reflectors. The average TS of rigid corner reflectors is 11.7 dB, while the average TS of elastic corner reflectors is only 1.9 dB.(2)Pattern width: The pattern width of a rigid corner reflector is significantly larger than that of an elastic corner reflector. The echo pattern width of a rigid corner reflector is approximately 46°, while that of an elastic corner reflector is approximately 18°.(3)Frequency stability of TS: When the frequency of the incident wave increases, the TS of a rigid corner reflector slowly increases and tends to be stable, indicating good frequency stability of the TS. However, the TS of an elastic corner reflector fluctuates greatly with the frequency of the incident wave, indicating poor frequency stability of the TS.

## 4. Experimental Verification

To verify the accuracy of the simulation data calculation and analysis, it is necessary to conduct a pool verification experiment.

This experiment was carried out in a reverberation pool. After formulating and adopting a reasonable experimental scheme, two underwater eight—cell triangular corner reflectors and the corresponding equipment deployment devices were carefully designed and fabricated. Subsequently, a test was conducted on the acoustic scattering characteristics of the array composed of these two corner reflectors, and the test results were carefully compared with the simulation calculation results.

### 4.1. Experimental System Design

The experiment employs the direct measurement method, which requires no special instruments or equipment, meets the requirements, and is convenient. Figure 19 is the schematic diagram of the measurement information flow for underwater experiments. The transmitting system consists of a pulse signal source transmitter and a transmitting transducer, while the receiving system comprises a hydrophone, a data collector, and a computer. The instruments and equipment used in the experiment are: an integrated pulse signal source transmitter, a transmitting transducer, an RHSA-30 standard hydrophone, a KEMO VBF40 filter amplifier, an NI USB-6216 data collector, and a computer.

To measure the stable and reliable echo signal, measurements should be conducted in the far field relative to the target, meaning the measurement distance should be greater than $L^{2}/\lambda$, where *L* is the side length of the corner reflector and λ is the incident wavelength. Due to the small size of the underwater corner reflector, this condition is easily met, and since the distance between the target and the receiving device is relatively short, the underwater propagation loss of the acoustic wave can be neglected. The incident wave adopts a CW pulse signal. To obtain the steady-state TS of the target, it is required that the transmitted signal contains at least ten or so waves within one pulse width. In the experiment, the acoustic scattering characteristics of the corner reflector at different incident angles are measured, rotating every 3° and measuring the echo at each angle.

To isolate boundary interference from the non-anechoic pool, the pulse width should be as narrow as possible. Assuming the pulse width is ξ, the measurement distance is *s*, and the pool depth is *h*, and assuming the transducer, hydrophone, and corner reflector are suspended at the same horizontal position, with a depth of 1/2 of the pool depth, if it is necessary to separate the echo signal from the water surface reflection interference, the requirement for the pool depth is $h>\sqrt{{\left(s+c\xi \right)}^{2}-s^{2}}$.

The arrangement of the experimental setup is shown in Figure 20, ensuring that the acoustic centers of the three experimental devices are at the same depth. The direction in which the transmitting transducer emits sound waves is directly towards the acoustic center of the diagonal reflector, ensuring the correctness of the incident angle of the sound source.

Calculate the TS based on the voltage value of the acoustic signal received by the hydrophone:(16)$$TS=10\mathrm{lg}\frac{{\left|I_{r}\right|}_{r=1}}{I_{i}}=20\mathrm{lg}\frac{U_{b}}{U_{d}}+20\mathrm{lg}\frac{d_{2}}{d_{1}}+20\mathrm{lg}(d_{1}+d_{2})$$

In the formula, $U_{b}$ represents the reflected sound intensity at a distance of 1 m from the target’s acoustic center; $U_{d}$ represents the voltage value of the target echo signal; *d*_1_ and *d*_2_ represent the distances between the hydrophone and the transducer, the hydrophone and the corner reflector, respectively; and *d*_3_ is the distance between the reflector and the pool wall.

Considering the small size of the experimental pool, in order to minimize the influence of the pool boundary, two types of small-sized metal corner reflectors were fabricated. The single-grid triangular corner reflector has the side length of 0.5 m and the steel plate thickness of 2 mm, as shown in Figure 21. The eight-grid corner reflector has the side length of 0.3 m and the steel plate thickness of 6 mm, as shown in Figure 22. The experimental setup in the pool is shown in Figure 23.

### 4.2. Single-Grid Corner Reflector Experiment

The acoustic TS measurement was conducted on the single-grid triangular metal corner reflector with acoustic incident angles of θ = 90° and φ = 0°~90°. The acoustic incident angle was changed by rotating the corner reflector with a rotation step size of 3°. The experimental data were organized and compared with the simulation results, as shown in Figure 24.

From Figure 24, the following can be seen:(1)There is a significant discrepancy between the test results and the simulation results of the single-grid rigid corner reflector in terms of numerical values and variation trends. The measured TS values are significantly lower than the simulated TS values of the rigid corner reflector. The average simulated TS value of the rigid corner reflector is 1.3 dB, while the average test TS value is −6.1 dB. The variation trends of the two are also inconsistent. Near the incident angle of 45°, the TS of the rigid corner reflector reaches maximum, while the test TS value reaches a minimum.(2)The test results are basically consistent with the variation trend of the simulation results for the single-grid elastic corner reflector. Both the simulated and test values of the TS of the elastic corner reflector reach their maximum values near the incident angles of 0~10° and 80~90°; they reach their minimum values near the incident angle of 45°, as well as near the incident angles of 10~20° and 70~80°. Due to the small size of the pool, reverberation from the pool walls, processing errors of the reflector, and measurement errors, there are certain deviations between the simulated data and the test data. The average value of the TS simulation is −4.8 dB, while the test average value is −6.1 dB.

### 4.3. Multi-Grid Corner Reflector Experiment

The acoustic TS measurement was conducted on the eight-grid metal circular corner reflector with sound wave incidence angles of θ = 90° and φ = 0°~90°. The sound wave incidence angle was changed by rotating the corner reflector with rotation step size of 3°. The experimental data were collated and compared with the simulation results, as shown in Figure 25.

From Figure 25, we can see the following:(1)There is a significant difference between the test results and the numerical simulation of the eight-grid rigid corner reflector. The test TS value is significantly lower than the simulated TS value of the rigid corner reflector. The average simulated TS value of the rigid corner reflector is −8.4 dB, while the average test TS value is −21.2 dB, difference of nearly 13 dB.(2)The test results are basically consistent with the simulation of the eight-grid elastic corner reflector in terms of numerical values and variation trends. The average value of the measured TS is −19.6 dB, while the average value of the simulation is −21.2 dB, with small difference. Near the incident angles of 0°, 90°, and 45°, both the simulated and test values of the elastic corner reflector’s TS reach their maximum values. Near the incident angles of 0~10° and 80~90°, both values reach their minimum values.(3)Due to the small side length of the corner reflector and the thinness of the flat plate, the actual TS value is very small, and the difference of TS compared to the rigid corner reflector becomes larger.

Two kinds of corner reflectors were used in the experiment. Compared with the eight-grid corner reflector, the flat plates of the single-grid corner reflector are very thin, which is not smooth due to deformation. The perpendicularity error between the plates of the single-grid corner reflector is larger. These reasons lead to the weakening of the acoustic echo of the single-grid corner reflector, and the actual measurement results of the TS become smaller. The experiment demonstrates that the preceding analysis of elastic loss is credible.

## 5. Conclusions

Based on the issue of acoustic echo elasticity loss of underwater corner reflectors, using underwater rigid corner reflectors as a benchmark, we compared the acoustic echo differences of similar types of underwater elastic corner reflectors. The acoustic echo elasticity loss of underwater corner reflectors was analyzed and experimental verification was conducted. The following specific conclusions were drawn:

(1) Whether it is the single-grid or multi-grid corner reflector, the actual acoustic echo characteristics of underwater corner reflectors conform to those of elastic bodies, which are significantly different from those of ideal rigid corner reflectors. Therefore, they should not be simplified as rigid bodies.

(2) The main manifestations of acoustic echo elasticity loss are reduced TS, narrowed directional pattern width, and deteriorated frequency stability of TS, which are detrimental to practical use.

(3) The thinner the plate constituting the corner reflector and the smaller its side length, the more pronounced the elasticity loss becomes, and the greater the disparity with the rigid corner reflector.

(4) Increasing the thickness of the corner reflector plate can reduce elastic loss, but the effect is limited. Even with steel plates up to 20 mm thick, the elastic loss remains significant, and the weight is large. Considering practical use, other technical measures need to be taken to suppress elastic loss.

## Figures and Tables

**Figure 1 sensors-25-03776-f001:**
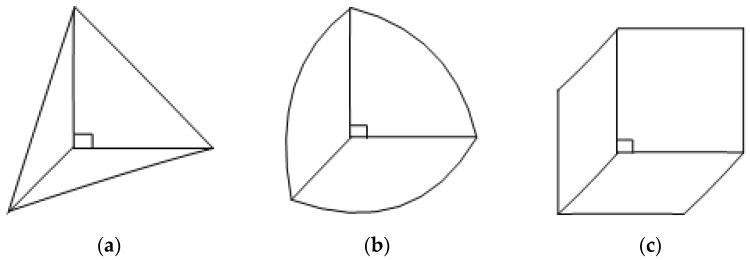
Single-grid corner reflector. (**a**) Three sided triangular corner reflector. (**b**) Three sided rounded corner reflector. (**c**) Three sided rectangular corner reflector.

**Figure 2 sensors-25-03776-f002:**
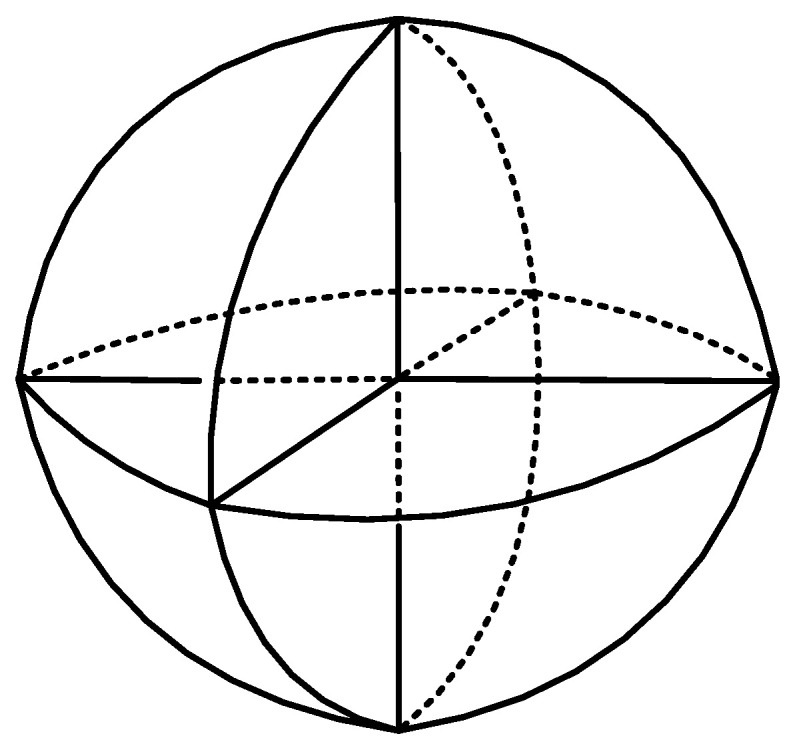
Schematic diagram of corner reflector linear array simulating submarine acoustic scattering.

**Figure 3 sensors-25-03776-f003:**
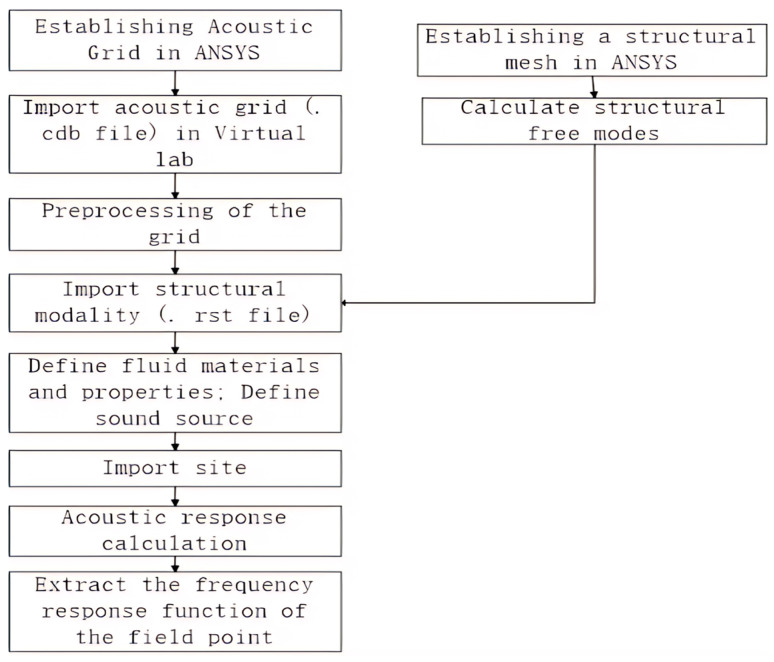
Simulation calculation process.

**Figure 4 sensors-25-03776-f004:**
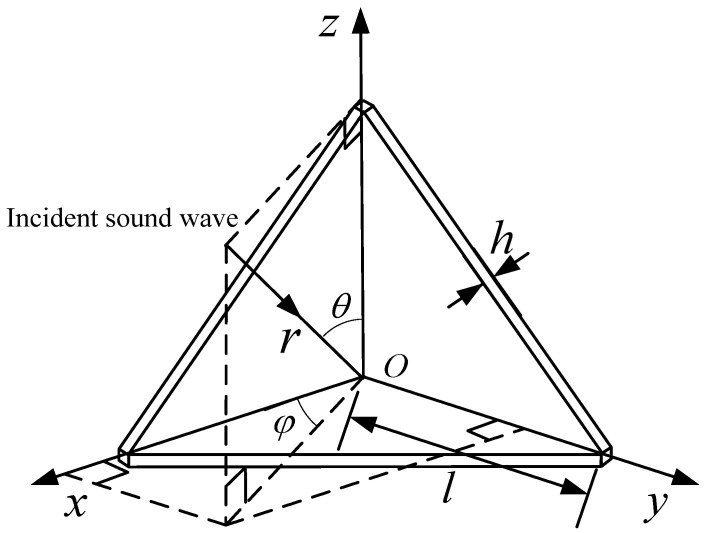
Schematic diagram of plane wave incident on single-grid corner reflector.

**Figure 5 sensors-25-03776-f005:**
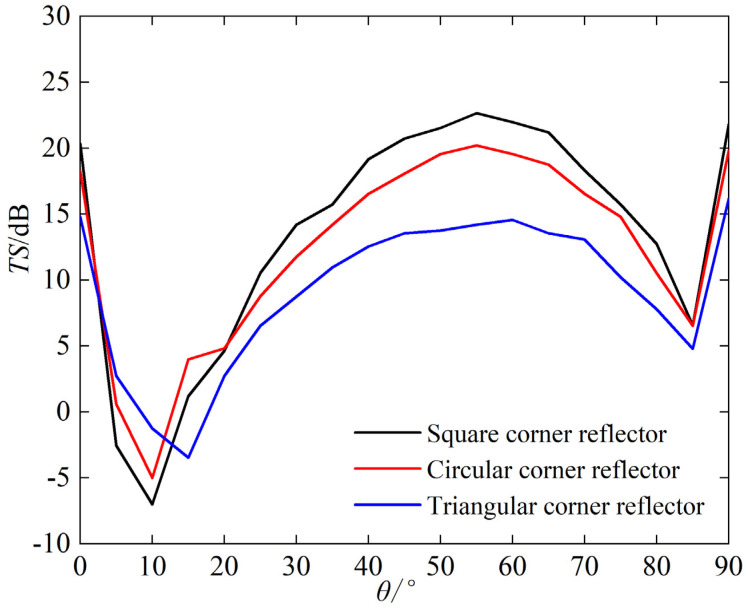
Target strength (TS) of *φ* = 45°, *θ* = 0~90°.

**Figure 6 sensors-25-03776-f006:**
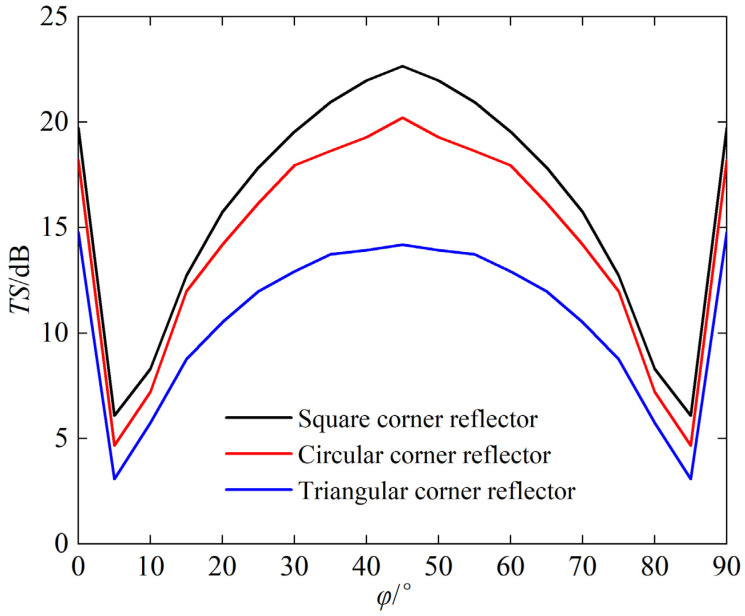
Target strength (TS) of *θ* = 55°, *φ* = 0~90°.

**Figure 7 sensors-25-03776-f007:**
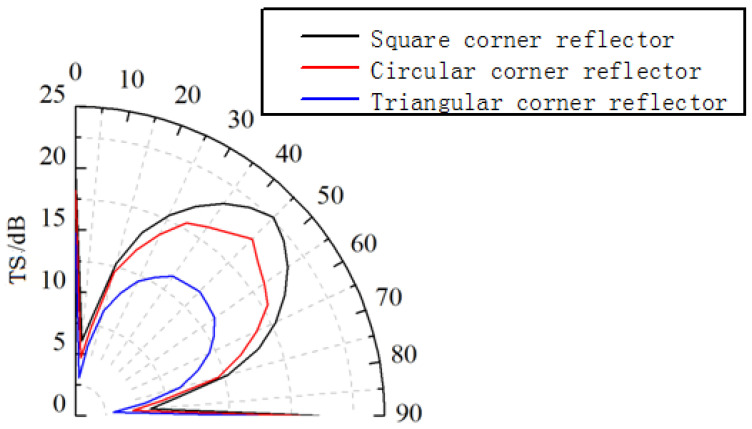
Scattering pattern of single-grid corner reflector.

**Figure 8 sensors-25-03776-f008:**
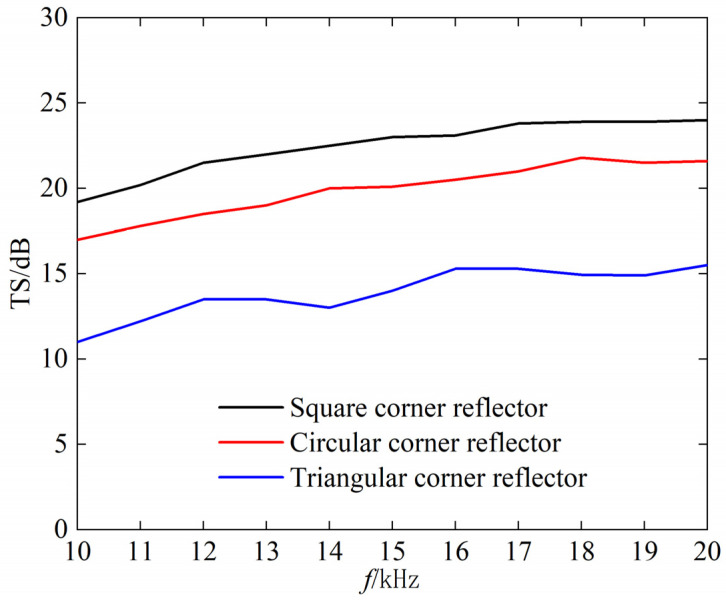
Relationship between target strength and frequency.

**Figure 9 sensors-25-03776-f009:**
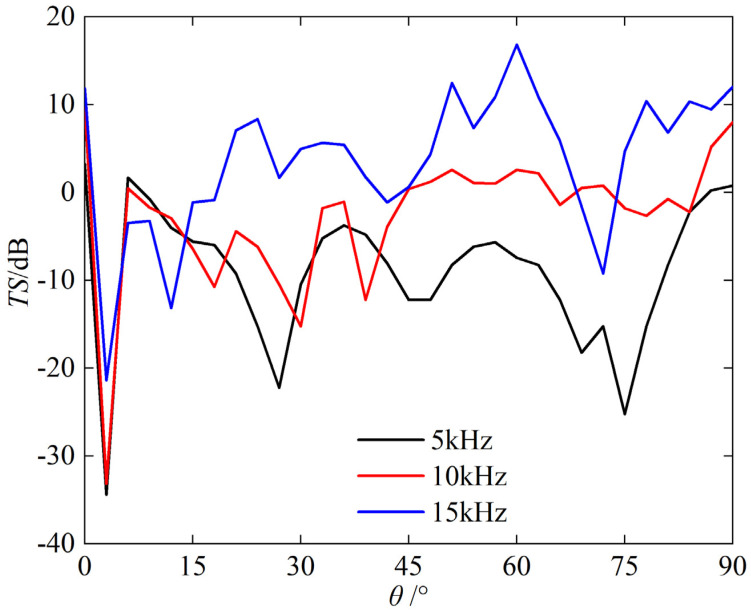
TS of *φ* = 45°, *θ* = 0~90°.

**Figure 10 sensors-25-03776-f010:**
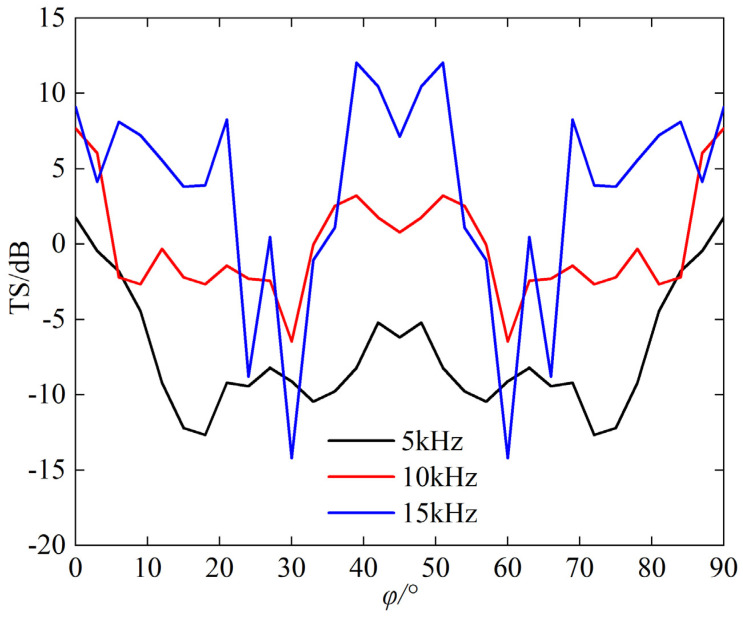
TS of *θ* = 55°, *φ* = 0~90°.

**Figure 11 sensors-25-03776-f011:**
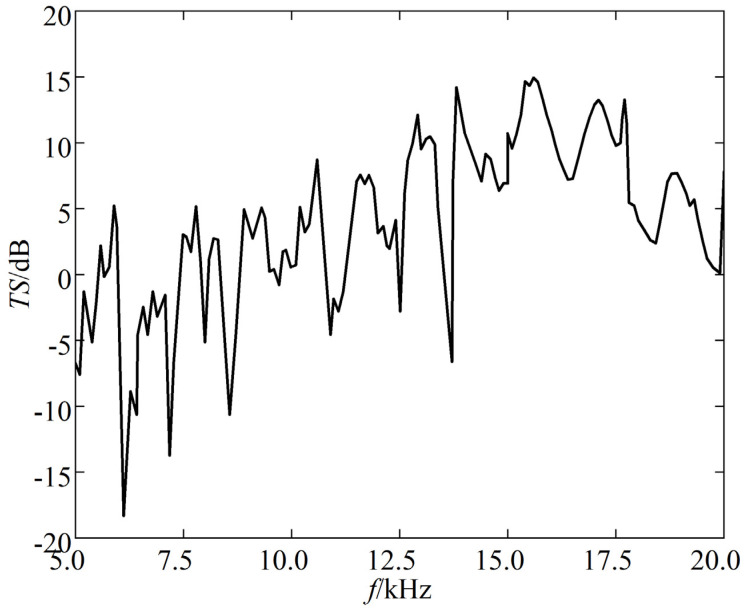
TS of triangular corner reflector.

**Figure 12 sensors-25-03776-f012:**
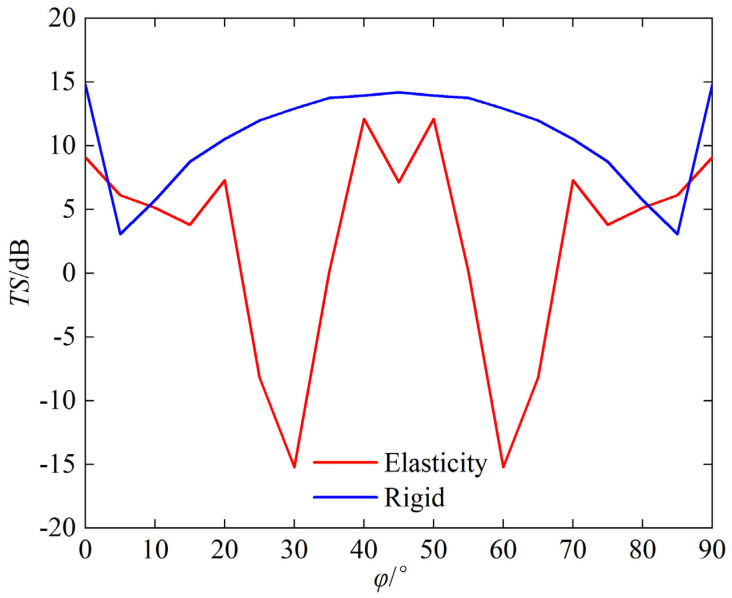
Comparison of TS with the change of φ.

**Figure 13 sensors-25-03776-f013:**
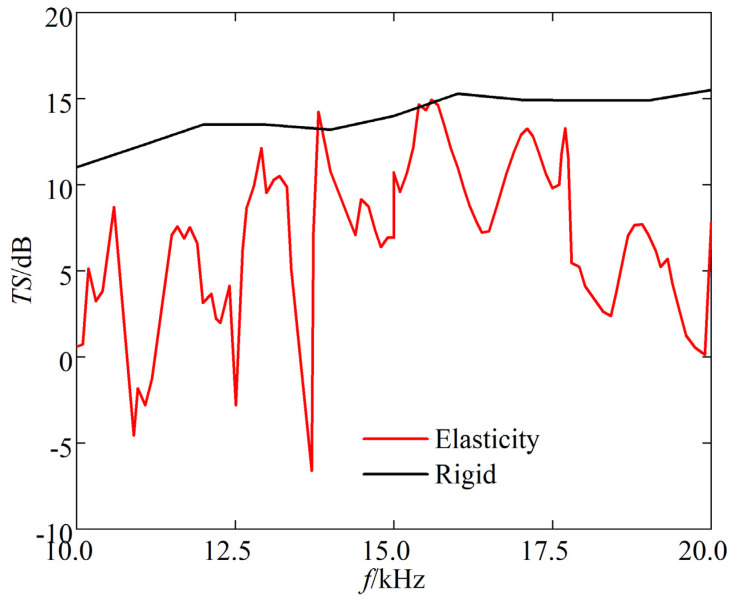
Comparison of TS with the change of *f*.

**Figure 14 sensors-25-03776-f014:**
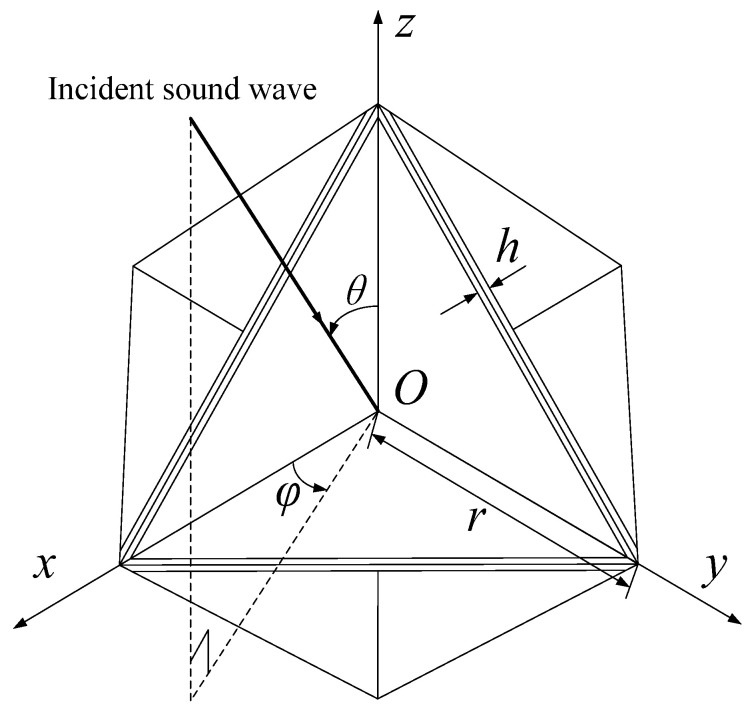
Schematic diagram of plane wave incident on eight-grid corner reflector.

**Figure 15 sensors-25-03776-f015:**
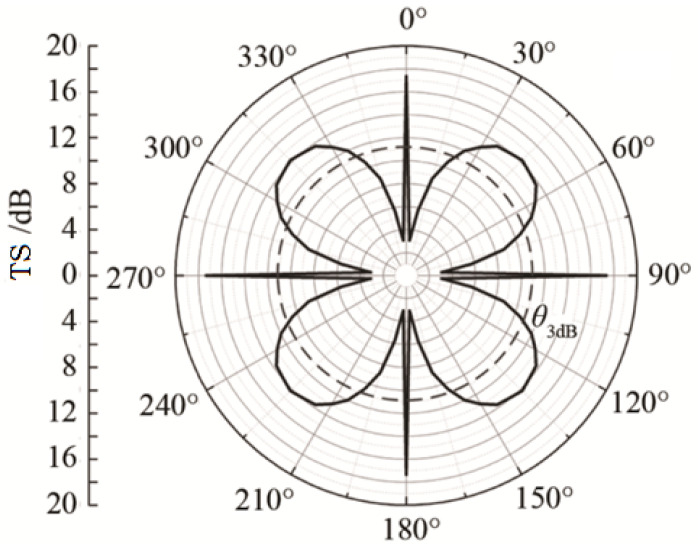
Scattering directional diagram of eight-grid corner.

**Figure 16 sensors-25-03776-f016:**
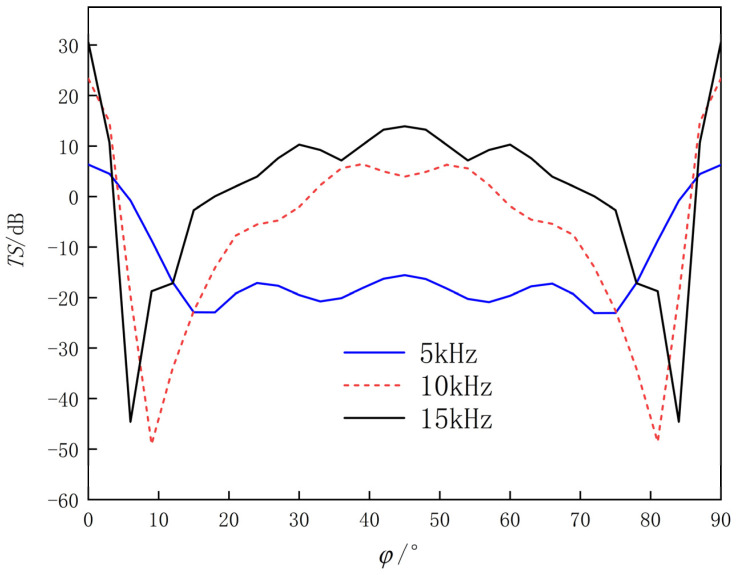
TS curve of eight-grid triangular corner reflector (θ
= 90°, φ = 0°~90°).

**Figure 17 sensors-25-03776-f017:**
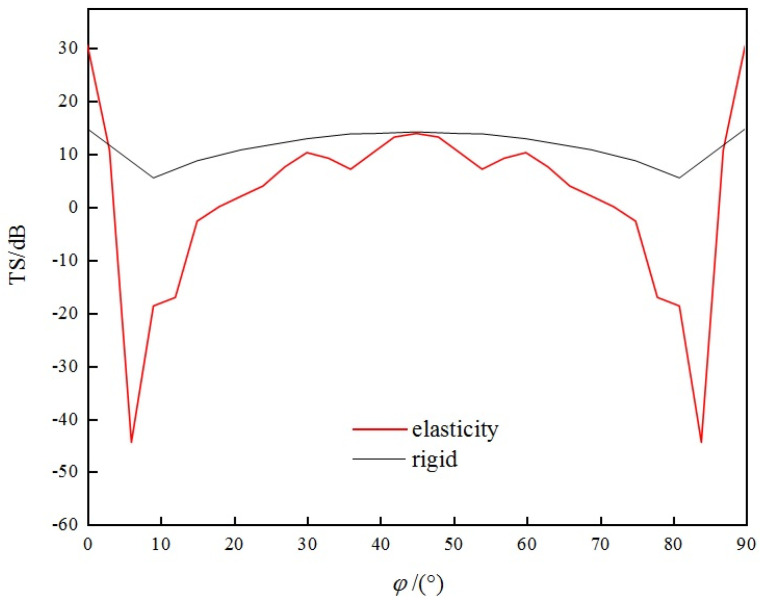
Comparison of TS with the change of *φ*.

**Figure 18 sensors-25-03776-f018:**
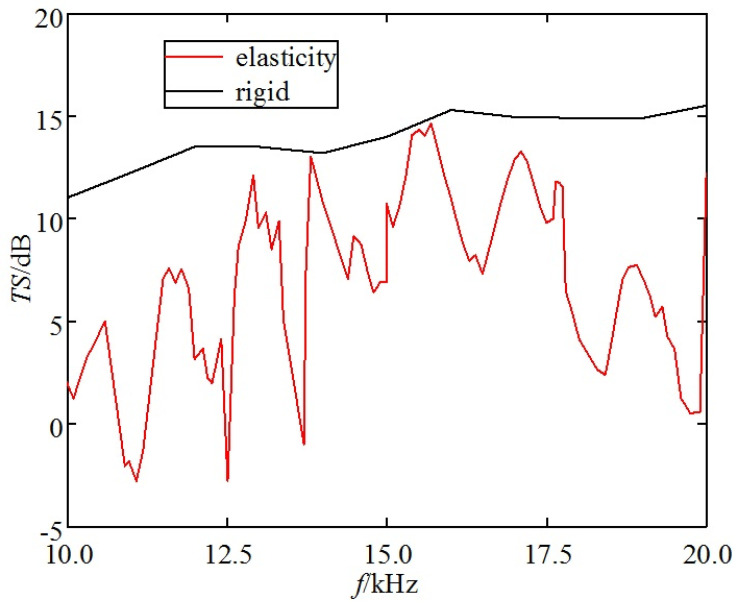
Comparison of TS with the change of *f*.

**Figure 19 sensors-25-03776-f019:**
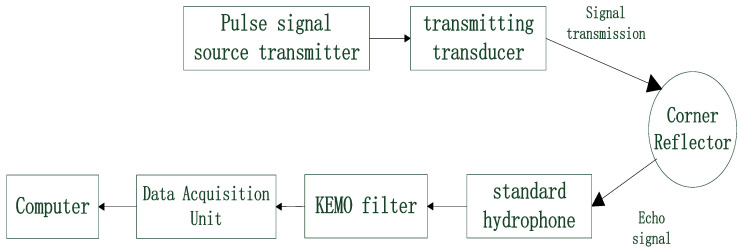
Schematic diagram of measurement system information flow.

**Figure 20 sensors-25-03776-f020:**
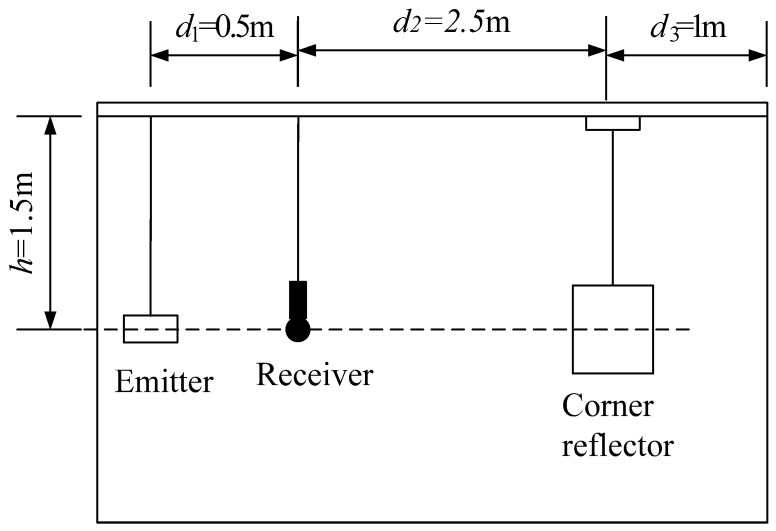
Schematic diagram of layout for trial.

**Figure 21 sensors-25-03776-f021:**
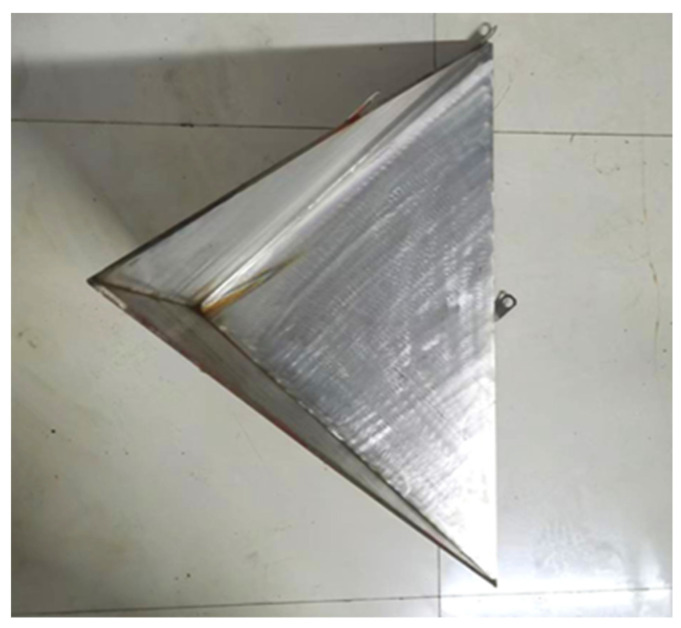
Single-grid steel plate corner reflector.

**Figure 22 sensors-25-03776-f022:**
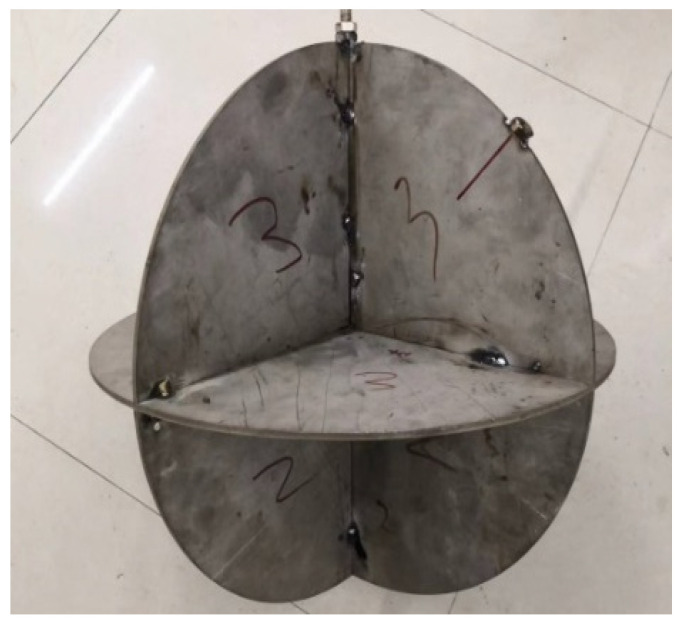
Eight-grid steel plate corner reflector.

**Figure 23 sensors-25-03776-f023:**
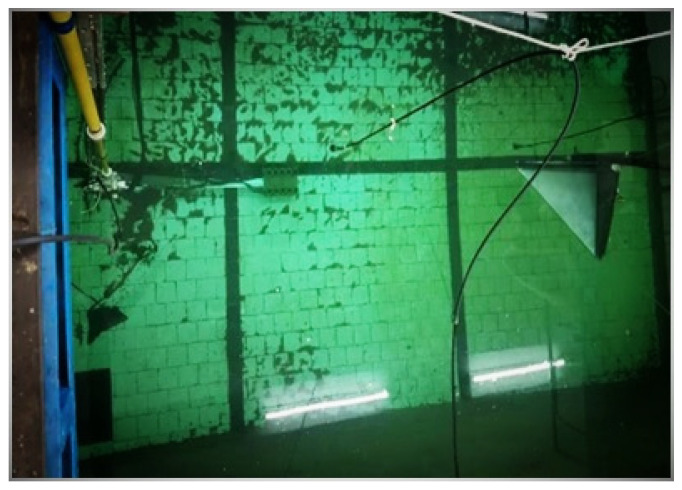
Site of experimental layout.

**Figure 24 sensors-25-03776-f024:**
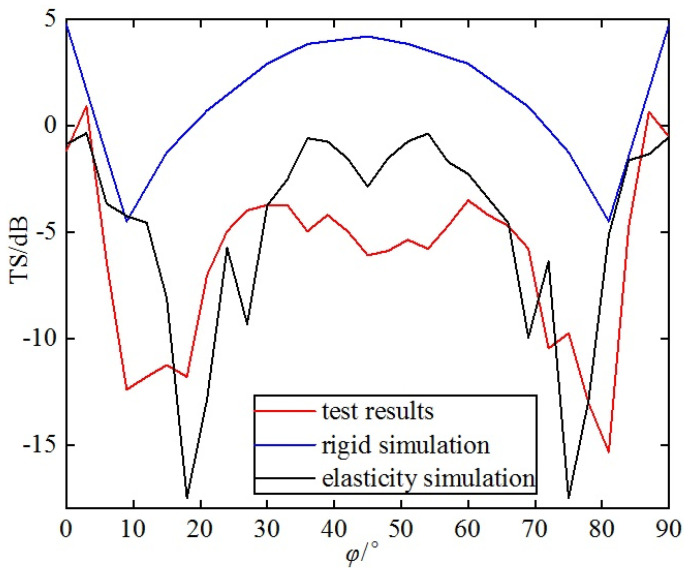
Comparison of results of single-grid steel plate corner reflector.

**Figure 25 sensors-25-03776-f025:**
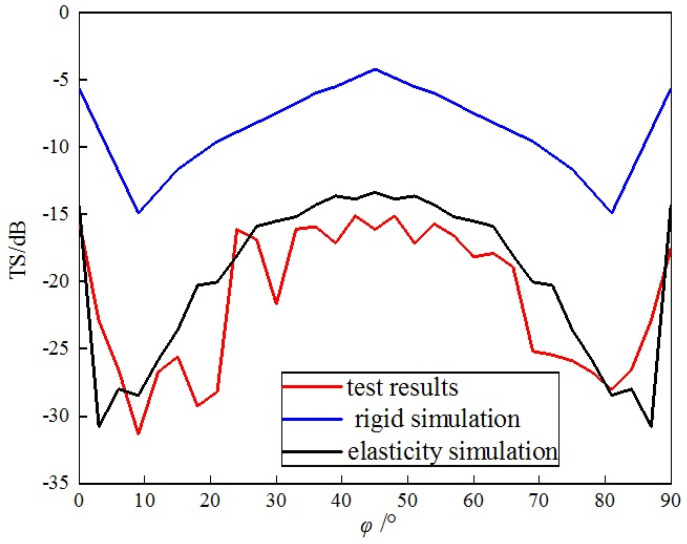
Experimental results of corner reflector array.

## Data Availability

The original contributions presented in this study are included in the article. Further inquiries can be directed to the corresponding authors.

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
