# Peer review of "Research on the Elastic Loss Characteristics of Acoustic Echoes from Underwater Corner Reflector"

_sensors, 2025, doi:10.3390/s25123776_

Round 1

Reviewer 1 Report

Comments and Suggestions for Authors

  1. The name of the corner reflector needs to be consistent throughout. For example, "Three sided triangular corner reflector" and "Triangular corner reflector", etc.
  2. The "Finite element method combined with direct boundary element method" introduced in Section 2.1 is a classic theory. In the article, only the calculation software based on these classic theories is used for modeling and calculation. The introduction of the classic theories can be deleted.
  3. The article studies the influence of the reflection characteristics of the elastic plate on the target strength of the corner reflector. Whether it is a single corner reflector or a combination of multiple ones has no bearing on the analysis of this influence. In the article, there are obvious differences in the error magnitudes between the experimental measurement results and the theoretical calculation results of the reflection of the two structures, but no reasonable explanation is given for such results.
  4. Figure 19 is unreasonable. The article measures the target echo, but the schematic diagram of the sound wave propagation process in the figure may mislead readers.
  5. The text format, font size, line format, and line thickness in the figures are rather chaotic.

Author Response

Comments 1: The name of the corner reflector needs to be consistent throughout. For example, "Three sided triangular corner reflector" and "Triangular corner reflector", etc.

Response 1: Thank you for your valuable comments. After rechecking the manuscript, the names of corner reflectors have been made consistent throughout. It should be noted that a corner reflector is a broad concept, and concepts such as "three-sided triangular corner reflector" and "three-sided circular corner reflector" are all subsets of corner reflectors.

Comments 2: The "Finite element method combined with direct boundary element method" introduced in Section 2.1 is a classic theory. In the article, only the calculation software based on these classic theories is used for modeling and calculation. The introduction of the classic theories can be deleted.

Response 2: Thank you for your valuable feedback. After discussion and verification, it was found that the introduction of classical theories had less innovation, so Section 2.1 has been deleted and the chapter arrangement has been reorganized.

Comments 3: The article studies the influence of the reflection characteristics of the elastic plate on the target strength of the corner reflector. Whether it is a single corner reflector or a combination of multiple ones has no bearing on the analysis of this influence. In the article, there are obvious differences in the error magnitudes between the experimental measurement results and the theoretical calculation results of the reflection of the two structures, but no reasonable explanation is given for such results.

Response 3: We have accepted the reviewers' comments. In the manuscript, the influence of the plate elasticity of corner reflectors on reflection characteristics is emphasized. The main reasons for the different errors in the two measurement results are explained as follows:

Two types of corner reflectors were used in the experiment. Compared with the multi-cell corner reflector, the single-cell corner reflector has thinner plates, and deformation causes unevenness; the perpendicularity error between the plates of the single-cell corner reflector is larger. These factors lead to weaker acoustic echo signals of the single-cell corner reflector and smaller actual measurement results of target strength.

Comments 4: Figure 19 is unreasonable. The article measures the target echo, but the schematic diagram of the sound wave propagation process in the figure may mislead readers.

Response 4: Thank you for your valuable input. The schematic diagram of Figure 19 has been revised, changing "target" in the figure to "corner reflector" to avoid misleading readers.

Comments 5: The text format, font size, line format, and line thickness in the figures are rather chaotic.

Response 5: Thank you for your valuable suggestions. All figures in the manuscript have been re-edited to achieve format consistency as much as possible.

We sincerely appreciate your valuable comments. All of the above-mentioned revisions have been completed in the manuscript. We would greatly appreciate it if you could kindly review and provide further feedback. Wishing you a pleasant life and smooth work!

Reviewer 2 Report

Comments and Suggestions for Authors

Comments and Suggestions for Authors are in "Review of sensors-3633234.pdf"

Author Response

Dear Reviewer, Thank you for your affirmation of the manuscript and the valuable suggestions provided. After a careful analysis of the issues and recommendations raised by you, we conducted an in-depth analysis and discussion on the manuscript content. Combining the comments from other reviewers, revisions have been made to the article, with modified sections distinguished in red. A detailed and substantial response to each of your comments is provided in the attached PDF.

Reviewer 3 Report

Comments and Suggestions for Authors

This paper conducts numerical simulations and experimental studies on the influence of structural elasticity on the acoustic target strength (TS) of corner reflectors, addressing a topic with clear engineering application value. The analysis of elastic effects is primarily approached from two perspectives: (1) comparison of TS values between elastic and rigid structures in numerical calculations, and (2) comparison between numerical results and experimental measurements. Overall, the authors have performed extensive comparative calculations and analyses for elastic corner reflector structures, along with valuable experimental work. However, the innovation appears somewhat limited, and the theoretical exploration of elastic mechanisms lacks depth, remaining largely at the level of comparative analysis. The accuracy of the computational results also requires further validation, as highlighted in the following aspects for discussion: 

  1. Well-Established Background:

   Corner reflector structures have been studied and applied for many years in both radar and underwater acoustics fields, leveraging their enhanced reflection properties due to geometric configurations. Numerous existing publications have investigated their geometric scattering mechanisms, multiple scattering effects, and spatial directivity of acoustic fields. 

  1. Consensus on Elastic Effects:

   It is widely recognized in the literature that structural elasticity generally degrades target strength. Consequently, some researchers have proposed designs such as double-layer steel plates with air backings for each reflector face to improve reflection efficiency by avoiding elastic losses. Others have suggested inflatable flexible structures as multi-faceted corner reflectors, utilizing total reflection from gas-filled surfaces to bypass the weakening effect of structural elasticity. 

  1. Overlap with Prior Work:

   The authors numerically analyze and compare single-grid triangular, square, and circular corner reflectors—a topic already addressed in Chen Wenjian’s work, which also delves into multiple scattering mechanisms and acoustic field hotspot characteristics. 

  1. Superficial Analysis of Elastic Mechanisms:

   The study relies heavily on comparative discussions of TS results under different conditions, which appears relatively simplistic and monotonous. A critical question remains: What specific elastic mechanisms are identified beyond these comparisons? 

  1. Validation Issues in Numerical Methods:

   The paper uses ANSYS for numerical simulations, with accuracy validation primarily based on experimental comparisons (e.g., Figure 24). However, discrepancies exceeding 10 dB are observed at many angles. While precise experimental measurements are challenging, the numerical method itself should first be validated against theoretical benchmarks to ensure reliability. Otherwise, the extensive numerical results lack robust support. 

   COMSOL, a widely adopted software for vibroacoustic coupling, could serve as an alternative. Many published studies on elastic structure scattering have validated COMSOL results against theoretical solutions, making it a viable option for cross-verification.  

Author Response

Dear Reviewer, Thank you for your valuable suggestions. After carefully reviewing your comments and considering the content of the manuscript, we have prepared a point-by-point response to each of your observations. The detailed replies are provided in the attached file.

Round 2

Reviewer 1 Report

Comments and Suggestions for Authors

Figure 19 still hasn't been revised reasonably. The article studies the echo characteristics of the target, and the experiment also measures the target's echo, that is, the target's backscattered wave. However, the sound wave propagation process in the figure shows the scattered waves in other directions, and this presentation is unreasonable.

Author Response

Dear Reviewer,   After re-examining your comments, we have reviewed the manuscript and Figure 19. In the original Figure 19, the wavy curves were used to illustrate the propagation process of sound waves in water, while the arrow directions represent the echoes of the measured targets in the experiment. The original intention of the wavy curves was to indicate that sound waves in water involve multiple paths, including reflection, scattering, and diffraction. However, as your prompt and analysis revealed, since the manuscript aims to study reflection, the dashed curves indeed had a misleading effect. We have revised and improved this part (please refer to the manuscript for details).   Thank you again for your valuable comments.

Reviewer 3 Report

Comments and Suggestions for Authors

The authors are requested to clearly specify the changes made in the manuscript corresponding to each of the reviewers' concerns.

Author Response

 Dear Reviewer,

After carefully analyzing your valuable comments and combining them with the previous round of manuscript revisions, we sincerely apologize for the fact that in the previous revision, due to significant overlaps among the comments from the three reviewers, we only highlighted the revised content in red font without specifically indicating the responses to each reviewer's comments. In the latest revised version of the manuscript, modifications addressing each reviewer's questions are marked using the annotation mode to facilitate your review. Additionally, we have provided supplementary responses to the second-round comments in an effort to maximize the improvement of the manuscript's quality.

Finally, thank you again for your precious feedback.

Round 3

Reviewer 3 Report

Comments and Suggestions for Authors

The authors have addressed all the reviewers' comments and suggestions.